# Optimization Studies and Compositional Oil Analysis of Pequi (*Caryocar brasiliense Cambess*) Almonds by Supercritical CO_2_ Extraction

**DOI:** 10.3390/molecules28031030

**Published:** 2023-01-19

**Authors:** Livia Silva Mateus, Juliete Martins Dutra, Rogério Favareto, Edson Antônio da Silva, Leandro Ferreira Pinto, Camila da Silva, Lucio Cardozo-Filho

**Affiliations:** 1Instituto Federal de Educação, Ciência e Tecnologia Goiano, Campus Rio Verde, Rio Verde 75901-970, GO, Brazil; 2Centro de Engenharias e Ciências Exatas, Universidade Estadual do Oeste do Paraná, Toledo 85903-000, PR, Brazil; 3Campus of Rosana, Sao Paulo State University (UNESP), Rosana 19274-000, SP, Brazil; 4Programa de Pós-graduação em Engenharia Química, Universidade Estadual de Maringá, Maringá 87020-900, PR, Brazil

**Keywords:** pequi, supercritical extraction, fatty acids, bioactive compounds, squalene

## Abstract

*Caryocar brasiliense* Cambess (pequi) is the fruit of the *pequizeiro* tree found in the Brazilian Cerrado (savanna). Supercritical fluids have been used to effectively extract bioactive chemicals. In light of the paucity of research on the supercritical extraction of pequi, in this study, experimental tests were conducted on the extraction of pequi almond oil using supercritical CO_2;_ the optimal extraction conditions were determined, and the fatty acids and active compounds in the oil were characterized. The experiments were conducted using the Box–Behnken experimental design of a three-variable system: pressure (15, 20, and 25 MPa), temperature (303.15, 318.15, and 333.15 K), and flow rate (2, 3, and 5 g.min^−1^). The optimal extraction conditions were 318.15 K, 25 MPa, and 5.0 g.min^−1^, which yielded 27.6 wt% of oil. The experimental kinetic curves were described using a second-order quadratic model (based on the Sovová model), which demonstrated a satisfactory correspondence with the kinetic curves. Significant amounts of squalene, stigmasterol, oleic fatty acids, and palmitic fatty acids were detected in pequi almond oil.

## 1. Introduction

*Caryocar brasiliense* Cambess, commonly known as pequi, is a fruit of the pequizeiro tree that grows abundantly in Brazil’s Cerrado (savanna) [1,2]. Pequizeiro is a leafy tree that can reach a height of 10 m, with twisted medium-sized stems and wide leaves [3]. Its fruits have a greenish epicarp (peel), yellowish mesocarp (pulp), brownish endocarp (thorny), and creamy chestnut hue [3]. Pequi is an excellent source of nutrients because it contains proteins, lipids, dietary fibers, carbohydrates, vitamins, carotenoids, and phenolic compounds [4,5,6]. Pequi almonds contain approximately 50% oil, which may be utilized to produce food, cosmetics, chemicals, and biofuels [1,2,7,8]. Rabbers et al. [9] reported that pequi oil is high in unsaturated fatty acids and possesses anti-inflammatory, antibacterial, antifungal, curative, antioxidant, and antitumor properties [10].

Studies have shown that pequi oil can be used in conjunction with physiotherapy to treat inflammation, lung infections, muscular discomfort, rheumatoid arthritis, and bruises [11]. Thus, research on the extraction of pequi oil is important because of its significant health benefits. Pequi oil is also a good source of squalene for human consumption, as well as for use in medicines and therapy. In addition, pequi oil contains active chemicals, such as stigmasterol, γ-tocopherol, α-tocopherol, and β-sistosterol [12].

According to the literature, the most frequently used methods for extracting oil from pequi almonds include solvent extraction [13], mechanical processing [14], and enzymatic extraction [10]. Pressurized/supercritical gas technology can also be used to extract oil [15,16]. This technique yields more oil compared to other techniques because of the intrinsic characteristics of supercritical fluid extraction, such as enhanced transport and diffusion of supercritical or near-supercritical gases into the matrix during the extraction process. Temperature and pressure control can also be used to extract specific bioactive components from the oleaginous matrix. Apart from the substantially higher yield, another advantage of supercritical extraction is the absence of residual organic solvents.

The use of carbon dioxide (CO_2_) eliminates the need for extensive heating while extracting thermolabile chemicals. Numerous studies have reported the extraction of oils and their active components using high-pressure gases (CO_2_ and propane) [17,18].

To address the lack of comparative research in the literature, this study investigates the influence of temperature and pressure on oil extraction from pequi almonds. Pressurized CO_2_ is used, focusing on improving the total yield and fatty acid profile and increasing the bioactive compound content. A second-order quadratic model is used to compute the mass transfers and characterize the observed kinetic curves.

## 2. Results and Discussion

The experimental conditions and total yields of pequi almond oil extracted using supercritical CO_2_ are listed in Table 1. The highest extraction yield of 27.6 wt% was obtained at 318.15 K, 25 MPa, and 5.0 g.min^−1^.

The yield in supercritical extraction may be associated with the duration of the experimental tests, as the kinetic curves did not present a maximum plateau. Figure 1 shows that there was no change in the inclination of the extraction kinetic curves. The changes in inclination were attributed to variations in the convective mass transfer mechanism. The mass transfer velocity was significantly affected by the convection mechanism in the fluid phase and relatively less by the diffusive mechanism. The gradual removal of lipid materials resulted in a discontinuity in the surface layer. At the beginning of this discontinuity, the extraction rate, governed by the diffusive mechanism, decreased. However, for pequi oil extraction, the extraction rate did not decrease despite variations in the CO_2_ flow rate. Thus, it appears that the duration of the experimental extraction was insufficient to reach the extractive equilibrium region. The characteristics of the extraction kinetics were similar for all the studied flows.

In general, increasing the CO_2_ flow in the extractor bed facilitated oil extraction. The highest extraction yields were obtained at a flow rate of 5 g.min^−1^. However, being sensitive to changes in pressure and temperature, these yields fluctuated significantly. In contrast, at a flow rate of 3.5 g.min^−1^, the extraction yields exhibited smaller fluctuations with variations in pressure and temperature. Given that pressure is a key factor in obtaining high yields, high CO_2_ densities help extract high oil yields.

Pressure also reduces the distance between molecules and enhances the interactions between CO_2_ and the sample, which promotes convective mass transfer [19]. Shi et al. [20] and Wang et al. [19] found that increasing the temperature and pressure helped remove carotenoids from pumpkin seeds via supercritical CO_2_ extraction.

These yield variations were verified by statistical analysis (Table 2). The results showed a linear relationship between temperature and pressure (Equation (1)), where T, P, and F correspond to temperature, pressure, and solvent flow rate, respectively.
Yield = 11.53 − 2.91 T − 7.34 P + 4.94 F + 0.89 T.P − 1.66 T.F + 3.25 P.F(1)

The coefficients of determination (R^2^) and adjusted R^2^ (R^2^adj) for the model were 0.994 and 0.989, respectively. The F-value = 218 and *p*-value < 0.05 indicate the significance of the model. Thus, the temperature, pressure, and flow rate had synergistic effects, i.e., an increase in these parameters led to the highest oil yields (Figure 2).

According to Table 3, the experimental parameters for the application of the mathematical model are as follows: bed density (ρ_bed_), porosity (*ε*), and apparent solubility (*C*_eq_). The model had only one adjustable parameter, *k =* 19.536 cm^3^.g^−1^.min^−1^. An AARD of 7.3% indicates excellent agreement between the experimental data and model values. Klein et al. (2020) [21] employed a second-order kinetic model to describe the supercritical extraction of *Eugenia pyriformis* Cambess leaves. They obtained *k* values in the range of 4–16 cm^3^.g^−1^.min^−1^, which are of the same order of magnitude as the values estimated in this study. Parameter *k* was used to generate the calculated extraction kinetics curves, as shown in Figure 1.

Analysis of fatty acids in pequi almond oil showed the following composition: oleic (~54%), palmitic (~37%), linoleic (~5%), and stearic (~1.5%) acids. Small amounts of myristic, palmitoleic, arachidic, linolenic, and gadoleic fatty acids were also observed (Table 4). The fatty acid profile obtained in this study was similar to that obtained by Soxhlet extraction methods reported in the literature [22,23]. The predominance of oleic acid, with no significant differences across extractions, indicates that the supercritical CO_2_ extraction process does not damage the primary fatty acids present in the oil.

The bioactive compounds detected in the extracts of pequi almonds were squalene, stigmasterol, β-sistosterol, octacosanol, triacontanol, γ-tocopherol, and α-tocopherol. Among these, squalene had the highest concentrations of 3069–14,220 mg per 100 g of oil. The squalene content extracted with pressurized CO_2_ is listed in Table 5.

The main fatty acids observed were oleic and palmitic acids. Under the optimal conditions, oleic acid in test 10 had the best extraction response, with a 6% statistical difference from that of the other tests. Palmitic acid exhibited the highest extraction in test 6 with 40% extraction. The differences between the experimental and quantitative extraction and identification results are attributed to the molecular mass of the fatty acids and to the temperature and pressure used in the supercritical extraction system [24,25].

Different extraction methods lead to variations in the fatty acid content. Although the supercritical extraction method is the most suitable, it is important to evaluate the extraction conditions, large-scale industrial processes, and industrial plants, which are influenced by the raw material and economic factors. In comparison, the Soxhlet extraction method remains the most widely applied process, although quantitative variations and degradation of molecules during heating limit further studies [26,27,28].

The highest solubility between CO_2_ and squalene was obtained at approximately 15 MPa, 303.15 K, and 3.5 g.min^−1^. However, despite the mass of squalene/mass of oil obtained under these conditions, the amount of oil extracted under the same experimental conditions was significantly low (1.1 wt%). Thus, these conditions were not optimal when the mass of the oil and squalene binary system was considered.

With regard to the quantitative variations between the tests, it should be noted that squalene, γ-tocopherol, α-tocopherol, stigmasterol, and β-sitosterol molecules promote numerous biological activities, such as hormone repositories, vitamin complexes, and cytotoxic agents in specific groups of cancer cell lines [29,30]. Quantitative analysis showed differences between the tests applied; however, in the test with high significance (run 2), there was a greater extraction per partition.

Among all extraction methods (e.g., hot or cold solvent extraction and crushing), the supercritical CO_2_ extraction method has the highest yield. However, the extraction plants still have to deal with problems such as low laboratory-scale production [31,32]. Although the supercritical system remains a small-scale system, the use of residues, mainly those from the epicarp and mesocarp of pequi fruit, has gained increasing attention because of the economic potential of the bioactive compounds contained in this fruit [33,34].

## 3. Materials and Methods

### 3.1. Sample Preparation

The pequi almonds were donated by farmers in Rio Verde-GO (17°47′52″ S, 50°55′40″ W) in Brazil. Prior to the experiments, the in natura samples were manually peeled and dried in a vented oven at 313.15 K for 24 h. The samples were shredded in a knife mill (Solab SL30, Brazil), sieved using a set of Tyler series (WS Tyler) sieves, and separated using a 20–42 mesh grain size. The true density (*ρ_t_*) of the particles was determined using helium gas pycnometry (Quantachrome Ultrapyc 1200 e, Germany). The samples were vacuum-packed in polyethylene bags and frozen.

### 3.2. Supercritical CO_2_ Extraction

The experiments were conducted using a CO_2_ cylinder, two thermostatically regulated baths, an Isco 500D syringe pump, a jacketed extraction vessel (1.91 cm in diameter and 16.8 cm in height), and an absolute pressure transducer (Smar LD301, Brazil) (Appendix A). Additional information on the instruments and approaches used in the experiment is available in previous studies [15,16,18,35,36].

The extractor was supplied with 0.01 kg of dried pequi almonds, while the remaining space in the extraction cell was filled with glass spheres (inert bed). As a result, the CO_2_ delivered to the extractor passed through the inert bed before reaching the ground seed. After reaching the appropriate extraction temperature, both the pump and extractor were simultaneously pressurized. After reaching the working pressure, the solution was allowed to cool for 30 min to achieve equilibrium and ensure that the solvent was saturated prior to the beginning of the extraction. The experiments were conducted using the Box–Behnken experimental design of a three-variable system: pressure (15, 20, and 25 MPa), temperature (303.15, 318.15, and 333.15 K), and flow rate (2, 3, and 5. and 5 g.min^−1^) (see Table 6). The CO_2_ (99.5% purity, Linde gas, Brazil) was compressed through the syringe pump at 293.15 K. For 110 min, the flow rate of the solvent was 3.5 g.min^−1^, regulated by a micrometric valve (Parker Autoclave Engineers, Erie, PA, USA), and maintained at 353.15 K. The total lipids extracted were collected in pre-weighed glass flasks at 10 min intervals (total of 110 min), and the extraction yield was measured gravimetrically using an analytical balance. The ultimate yield was determined by dividing the total mass extracted by the initial mass of leaves in the extractor (dry basis). The analyses were performed in a random order.

### 3.3. Kinetics of the Pequi Almond Oil Extraction

The second-order model, modified by De Sousa et al. [37], assumes that the extraction rate is proportional to both the residual capacity of the solvent for solute extraction and solute concentration in the solid matrix. The model is described by Equations (2) and (3), respectively.
(2)$$\frac{\partial C}{\partial t}+\frac{\rho_{\mathrm{bed}}}{\varepsilon }\frac{\partial q}{\partial t}+u\frac{\partial C}{\partial z}=0$$
(3)$$\frac{\partial q}{\partial t}=-\mathrm{kq}\left(C_{\mathrm{eq}}-C\right)$$
where *C* is the solute concentration in the solvent (kg m^−3^), *t* is the extraction time (min), *ρ_bed_* is the bed density (kg m^−3^), *ε* is the porosity, *q* is the solute concentration in the solid matrix (kg m^−3^), *u* is the interstitial velocity (m s^−1^), *z* is the axial coordinate of the bed, *k* is the kinetic constant (m^3^ kg^−1^ min^−1^), and *C_eq_* is the equilibrium solute concentration of the solvent (kg m^−3^).

For the second-order model, the initial and boundary conditions are described by Equations (4)–(6).
(4)$$C\left(0,z\right)=C_{\mathrm{eq}}$$
(5)$$q\left(0,z\right)=q_{0}$$
(6)$$C\left(t,0\right)=\{\begin{array}{c} C_{\mathrm{eq}}\;t=0\\ 0\;t>0 \end{array}$$
where q_0_ is the initial solute concentration in the solid matrix (kg m^−3^).

Equation (7) describes the analytical solution of the second-order model.
(7)$$\frac{C}{C_{\mathrm{eq}}}=\{\begin{array}{c} 1\;t<t_{r}\\ 1-\frac{1}{\left(e^{A}+e^{-B}-1\right)e^{B}}\;t>t_{r} \end{array}$$
where A=(z/u)β; B=[(−tu+z)β]/αu; $\beta ={\mathrm{kC}}_{\mathrm{eq}}\alpha$; $\alpha =\left(\rho_{b}q_{0}\right)/\left({\varepsilon C}_{\mathrm{eq}}\right)$, and $t_{r}$ is the residence time of the solvent in the bed.

The total mass extracted is obtained by Equation (8).
(8)$$m_{\mathrm{ext}}=\int_{0}^{t_{e}}C_{z=H}\mathrm{wt}=\{\begin{array}{c} C_{\mathrm{eq}}wt\;t<t_{r}\\ C_{\mathrm{eq}}wt-\frac{C_{\mathrm{eq}}w\alpha }{\beta }ln\left(e^{\frac{\left(z\beta \right)}{u}}+e^{\frac{\left[-\left(\mathrm{tu}+z\right)\beta \right]}{\alpha u}}-1\right)\;t>t_{r} \end{array}$$
where $C_{z=H}$ is the solute concentration in the solvent at the extractor outlet (kg.m^−3^); H is the length of the extractor (m); and w is the volumetric flow rate of the solvent (m^3^.min^−1^). In the second-order kinetic model, only one adjustable parameter (k) exists. A quadratic interpolation algorithm was used to estimate the values of the parameter and the following objective function according to Equation (9):(9)$$F_{OBJ}=\sum_{j=1}^{N}{\left(\frac{m_{extract,j}^{Exp}-m_{extract,j}^{Calc}}{m_{extract,j}^{Exp}}\right)}^{2}$$

The goodness of fit of the model was quantified using the average absolute relative deviation (AARD) according to Equation (10).
(10)$$\mathrm{AARD}\left(\%\right)=\frac{100}{N}\sum_{j-1}^{N}\left|\frac{m_{extract,j}^{Exp}-m_{extract,j}^{Calc}}{m_{extract,j}^{Exp}}\right|$$
where *N* is the number of experimental data points from all extraction curves, and $m_{extract,j}^{Exp}$ and $m_{extract,j}^{Calc}$ are the experimental and calculated mass of the extract, respectively.

### 3.4. Oil Characterization

To determine the fatty acid profile, the samples were derivatized using an adapted version of the methodology described by Santos Júnior et al. [38], in which the samples were derivatized in methanol with sulfuric acid and potassium hydroxide, followed by chromatographic analysis (Shimadzu, GCMS-QP2010 SE), as described by Rodrigues et al. [39].

Gas chromatography (Shimadzu, GCMS-QP2010 SE) linked to mass spectrometry and equipped with a Shimadzu Rtx-5MS capillary column (30 m × 0.25 mm × 0.25 mm) was used to assess the active substances using a modified version of the approach published by Du and Ahn [40]. The sample was derivatized using 20 μL N,O-bis(trimethylsilyl)trifluoroacetamide)/trimethylsilyl chloride (333.15 K for 30 min), followed by the addition of 5-cholestane. Data were collected using the GCMS Postrun Analysis program (Shimadzu), which includes the databases for the NIST14.lb and NIST14.lbs spectral libraries.

### 3.5. Statistical Analysis

The data were subjected to analysis of variance (ANOVA) at 5% significance, followed by Tukey’s test. The main effects and interactions were computed using Design-Expert software version 12 [41]. Statistica software version 8.0 [42] was used to compute the main effects and interactions and to determine the influence of independent factors on the response.

## 4. Conclusions

The best pequi almond oil extraction conditions were pressure, temperature, and flow rate of 25 MPa, 318.15 K, and 5 g min^−1^, respectively, to obtain 27.6 wt% of oil. Oleic acid (>50 wt%) and palmitic acid (>35 wt%) were the most abundant fatty acids in the pequi almond oil. Fatty acid concentrations remained constant under all experimental conditions. The oil contained high amounts of squalene and stigmasterol. The second-order mathematical model effectively adjusted the extraction curves, contributing to the understanding of the extraction phases and mass transfer mechanisms.

## Figures and Tables

**Figure 1 molecules-28-01030-f001:**
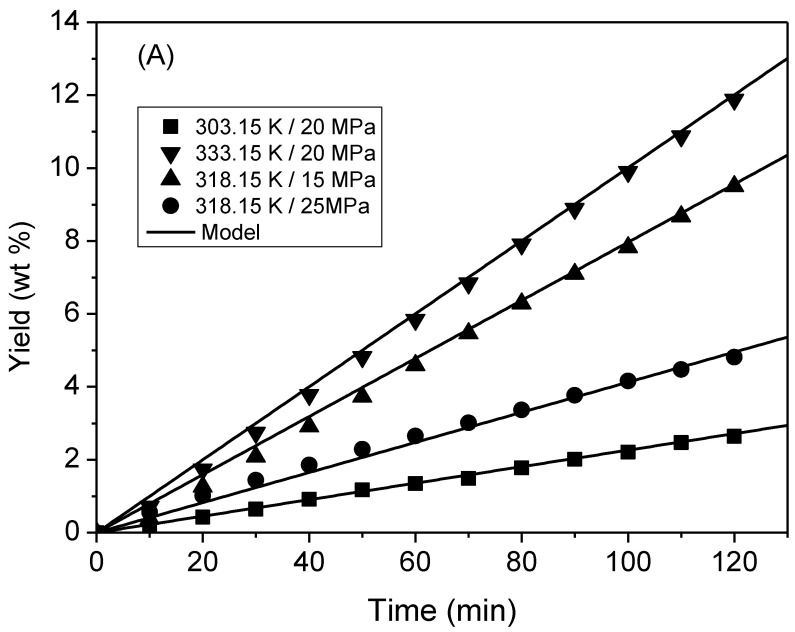
Experimental kinetic curves for pequi almond oil extraction with supercritical CO_2_ fitted using the second-order model. Flow rates of (**A**) 2.0 g.min^−1^, (**B**) 3.5 g.min^−1^, and (**C**) 5.0 g.min^−1^.

**Figure 2 molecules-28-01030-f002:**
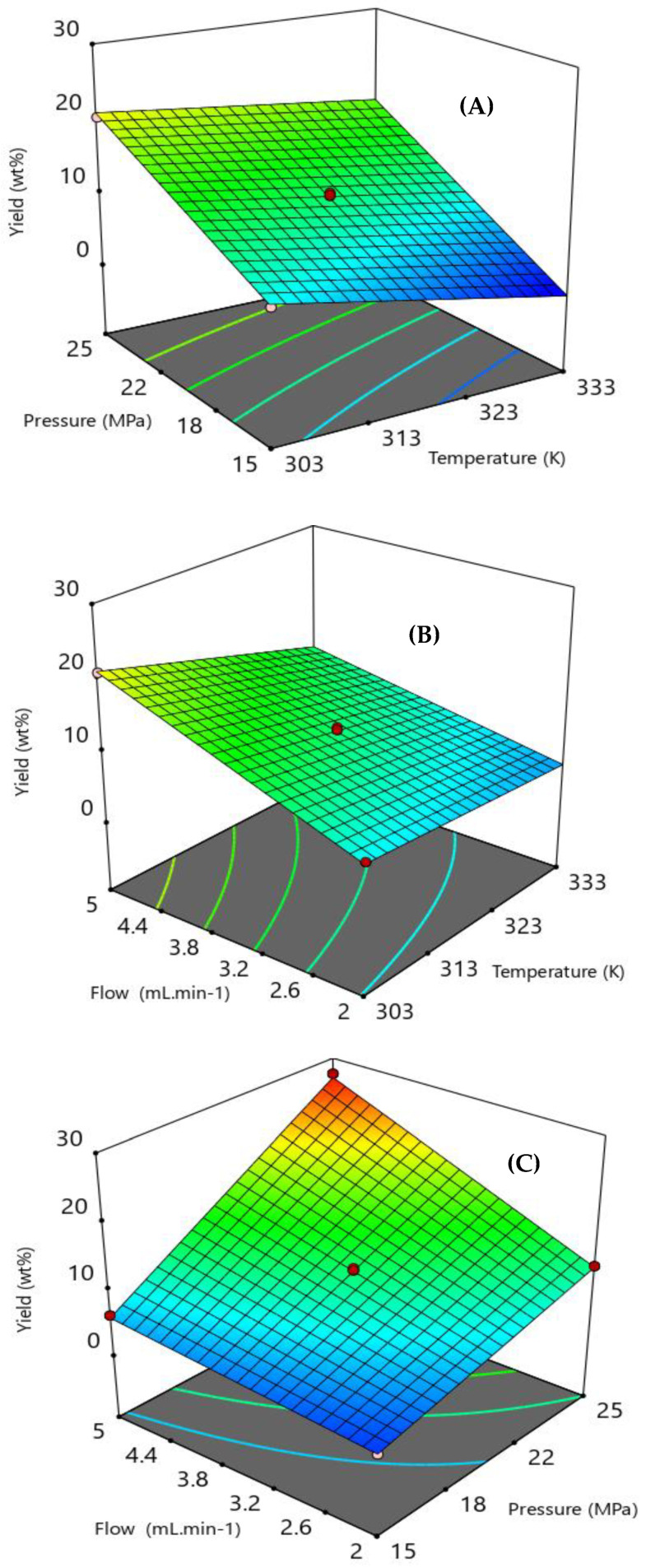
Response surface showing the behavior of the extraction yield as a function of (**A**) temperature and pressure with constant solvent flow rate of 3.5 g.min^−1^; (**B**) temperature and solvent flow rate with constant pressure of 20 MPa; and (**C**) pressure and solvent flow rate with constant temperature of 318.15 K.

**Table 1 molecules-28-01030-t001:** Experimental conditions and extraction yield results for pequi oil extraction using supercritical CO_2_.

Run	Temperature	Pressure	Flow Rate	Yield
	(K)	(MPa)	(g.min^−1^)	(wt%)
1	303.15	15	3.5	7.6
2	333.15	15	3.5	1.1
3	303.15	25	3.5	20.3
4	333.15	25	3.5	17.4
5	303.15	20	2.0	8.1
6	333.15	20	2.0	4.5
7	303.15	20	5.0	20.9
8	333.15	20	5.0	10.6
9	318.15	15	2.0	2.5
10	318.15	25	2.0	10.9
11	318.15	15	5.0	6.3
12	318.15	25	5.0	27.6
13	318.15	20	3.5	11.6
14	318.15	20	3.5	12.1
15	318.15	20	3.5	11.7

**Table 2 molecules-28-01030-t002:** Data on ANOVA for oil obtained using Box–Behnken experimental design of a three-variable system for CO_2_ extractions.

Terms	Sum of Squares	Degrees of Freedom	Mean Squares	F-Value	*p*-Value
Model	750.5	6	125.1	218	<0.0001
T	67.6	1	67.6	118	<0.0001
P	431.3	1	431.3	753	<0.0001
F	195.3	1	195.3	341	<0.0001
T.P	3.2	1	3.2	5.5	0.05
T.F	11.0	1	11.0	19.2	0.002
P.F	42.1	1	42.1	73.5	<0.0001
Residual	4.6	8	0.6		
Lack of Fit	4.5	6	0.7	13.6	0.07
Pure Error	0.1	2	0.06		
Cor Total	755.1	14			

T = Temperature; P = Pressure; and F = Flow rate.

**Table 3 molecules-28-01030-t003:** Experimental conditions, extraction yield results, and parameters of the mathematical model for pequi oil extraction using supercritical CO_2_.

Run	Temperature	Pressure	Flow Rate	C_eq_ × 1000 ^a^	Yield
	(K)	(MPa)	(g.min^−1^)	(goil.cm^−3^)	(wt%)
1	303.15	15	3.5	1.87	7.6
2	333.15	15	3.5	0.42	1.1
3	303.15	25	3.5	5.39	20.3
4	333.15	25	3.5	4.78	17.4
5	303.15	20	2.0	3.89	8.1
6	333.15	20	2.0	1.98	4.5
7	303.15	20	5.0	3.89	20.9
8	333.15	20	5.0	1.98	10.6
9	318.15	15	2.0	1.09	2.5
10	318.15	25	2.0	5.11	10.9
11	318.15	15	5.0	1.09	6.3
12	318.15	25	5.0	5.12	27.6
13–15 ^e^	318.15	20	3.5	3.03	11.8

^a^*C*_eq_, apparent solubility (g of oil per volume of solvent). ^e^ Mean ± standard deviation (n = 3). *ε*, porosity: 0.8125. ρ_bed_, density of the bed (kg m^−3^): 215.7.

**Table 4 molecules-28-01030-t004:** Fatty acid profile of pequi almond oils obtained by supercritical CO_2_.

Run	Fatty Acid (%) ^1^								
	Myristic	Palmitic	Palmitoleic	Stearic	Oleic	Linoleic	Arachidic	Linolenic	Gadoleic
1	0.35 ± 0.00 ^c^	37.76 ± 0.04 ^d.e^	0.41 ± 0.00 ^a.b^	1.48 ± 0.04 ^d^	53.66 ± 0.02 ^c^	6.09 ± 0.01 ^b^	0.08 ± 0.00 ^b.c^	0.13 ± 0.00 ^d^	0.05 ± 0.01 ^a^
2	0.41 ± 0.00 ^a^	35.18 ± 0.02 ^f^	0.43 ± 0.01 ^a^	1.56 ± 0.01 ^c.d^	54.07 ± 0.03 ^b^	6.035 ± 0.02 ^b^	0.01 ± 0.00 ^a^	0.15 ± 0.00 ^c^	0.05 ± 0.00 ^a^
3	0.31 ± 0.00 ^e^	37.31 ± 0.03 ^f^	0.36 ± 0.01 ^e.f^	1.76 ± 0.00 ^a.b.c^	54.23 ± 0.01 ^a.b^	5.69 ± 0.02 ^c.d^	0.08 ± 0.01 ^b^	0.18 ± 0.00 ^b^	0.06 ± 0.01 ^a^
4	0.27 ± 0.00 ^f^	37.35 ± 0.00 ^e.f^	0.32 ± 0.01 ^g^	1.92 ± 0.06 ^a.b^	54.48 ± 0.04 ^a^	5.32 ± 0.00 ^c.d^	0.07 ± 0.00 ^c.d^	0.21 ± 0.00 ^a^	0.05 ± 0.01 ^a^
5	0.33 ± 0.00 ^d^	38.05 ± 0.12 ^d^	0.39 ± 0.00 ^b.c.d^	1.70 ± 0.01 ^a.b.c.d^	53.58 ± 0.07 ^c^	5.66 ± 0.05 ^c.d^	0.08 ± 0.00 ^c^	0.15 ± 0.00 ^c^	0.04 ± 0.00 ^a^
6	0.37 ± 0.00 ^b^	40.56 ± 0.04 ^a^	0.37 ± 0.00 ^c.d.e.f^	1.95 ± 0.01 ^a^	51.36 ± 0.02 ^g^	5.10 ± 0.00 ^g^	0.06 ± 0.00 ^d^	0.17 ± 0.00 ^b^	0.04 ± 0.00 ^a^
7	0.31 ± 0.00 ^e^	38.16 ± 0.03 ^c.d^	0.37 ± 0.00 ^c.d.e.f^	1.83 ± 0.00 ^a.b^	53.24 ± 0.05 ^d.e^	5.78 ± 0.01 ^c^	0.07 ± 0.00 ^b.c.d^	0.17 ± 0.00 ^b^	0.05 ± 0.00 ^a^
8	0.33 ± 0.01 ^c.d^	38.80 ± 0.05 ^b^	0.37 ± 0.00 ^c.d.e.f^	1.69 ± 0.06 ^b.c.d^	52.87 ± 0.01 ^f^	5.64 ± 0.02 ^d.e^	0.07 ± 0.00 ^c.d^	0.15 ± 0.00 ^c^	0.05 ± 0.00 ^a^
9	0.32 ± 0.01 ^d.e^	37.45 ± 0.01 ^e.f^	0.37 ± 0.01 ^d.e.f^	1.56 ± 0.04 ^c.d^	54.26 ± 0.00 ^a.b^	5.77 ± 0.03 ^c^	0.07 ± 0.00 ^c.d^	0.15 ± 0.00 ^c^	0.04 ± 0.00 ^a^
10	0.33 ± 0.00 ^c^	0.39 ± 0.05 ^d^	0.39 ± 0.00 ^b.c.d.e^	1.18 ± 0.08 ^e^	53.45 ± 0.04 ^c.d^	6.25 ± 0.07 ^a^	0.08 ± 0.00 ^b.c^	0.15 ± 0.00 ^c^	0.05 ± 0.00 ^a^
11	0.35 ± 0.00 ^c^	38.56 ± 0.00 ^b.c^	0.40 ± 0.00 ^b.c^	1.73 ± 0.06 ^a.b.c.d^	52.99 ± 0.03 ^e.f^	5.77 ± 0.02 ^c.d^	0.07 ± 0.00 ^c.d^	0.15 ± 0.00 ^c^	0.04 ± 0.01 ^a^
12	0.27 ± 0.01 ^f^	37.13 ± 0.05 ^f^	0.35 ± 0.00 ^f.g^	1.92 ± 0.01 ^a.b^	54.48 ± 0.04 ^a^	5.52 ± 0.02 ^e^	0.07 ± 0.00 ^c.d^	0.21 ± 0.00 ^a^	0.05 ± 0.00 ^a^
13–15 ^a^	0.32 ± 0.01 ^d.e^	38.13 ± 0.35 ^d^	0.41 ± 0.00 ^a.b^	1.84 ± 0.18 ^a.b^	53.29 ± 0.22 ^d^	5.69 ± 0.04 ^c.d^	0.08 ± 0.00 ^b.c^	0.17 ± 0.01 ^b^	0.04 ± 0.00 ^a^

Mean ± standard deviation (n = 3) ^1^ in relative area. Means followed by the same letters (in each column) did not differ statistically (*p* > 0.05).

**Table 5 molecules-28-01030-t005:** Bioactive compounds of pequi almond oils obtained by supercritical CO_2_.

Run	Compound (mg per 100 g Oil)						
	Squalene	Y-Tocopherol	Octacosanol	α-Tocopherol	Stigmasterol	Triacontanol	β-Sitosterol
1	5032.05 ± 59.04 ^f^	9.17 ± 0.12 ^e^	27.59 ± 0.15 ^d^	11.92 ± 0.14 ^d.e^	133.92 ± 1.64 ^d^	12.99 ± 0.12 ^c^	35.96 ± 2.92 ^b.c^
2	14219.82 ± 68.39 ^a^	27.01 ± 1.11 ^a^	34.06 ± 0.47 ^c^	24.65 ± 0.77 ª	324.88 ± 2.07 ^a^	16.69 ± 0.86 ^b^	70.61 ± 1.35 ^a^
3	2972.47 ± 1.62 ^i^	9.26 ± 0.05 ^e^	9.89 ± 0.30 ^g^	9.25 ± 0.87 ^f.g.h^	90.48 ± 2.92 ^h^	5.56 ± 0.03 ^f^	25.68 ± 1.14 ^d.e.f^
4	3069.31 ± 24.20 ^i^	9.55 ± 0.11 ^e^	9.59 ± 0.41 ^g^	7.30 ± 1.39 ^h^	93.54 ± 4.05 ^g.h^	7.19 ± 0.21 ^e.f^	23.73 ± 0.12 ^d.e.f^
5	4225.7 ± 135.21 ^g^	12.04 ± 0.78 ^d^	20.73 ± 0.67 ^e^	9.99 ± 0.19 ^e.f.g^	119.50 ± 0.33 ^e^	10.36 ± 0.19 ^d^	24.05 ± 2.27 ^d.e.f^
6	7076.92 ± 45.31 ^c^	18.99 ± 0.01 ^b^	40.65 ± 0.44 ^b^	16.53 ± 0.39 ^b^	185.65 ± 0.37 ^c^	18.82 ± 1.03 ^b^	39.18 ± 2.52 ^b^
7	3057.3 ± 28.68 ^i^	8.95 ± 0.38 ^e^	14.08 ± 0.17 ^f^	8.54 ± 0.70 ^g.h^	88.60 ± 1.45 ^h^	7.83 ± 0.43 ^e^	20.80 ± 1.56 ^e.f^
8	5507.90 ± 50.45 ^e^	16.89 ± 0.35 ^c^	22.52 ± 0.42 ^e^	11.63 ± 0.22 ^d.e.f^	143.42 ± 0.44 ^d^	12.71 ± 0.54 ^c^	30.20 ± 0.17 ^c.d^
9	9021.06 ± 198.23 ^b^	18.59 ± 1.04 ^b.c^	36.91 ± 2.88 ^c^	14.80 ± 0.57 ^b.c^	197.97 ± 0.52 ^b^	16.85 ± 1.13 ^b^	38.18 ± 0.63 ^b^
10	3756.2 ± 81.36 ^h^	10.94 ± 0.26 ^d.e^	10.99 ± 0.27 ^f.g^	14.00 ± 0.79 ^c.d^	103.97 ± 1.99 ^f.g^	5.80 ± 0.21 ^e.f^	20.60 ± 0.77 ^f^
11	6671.90 ± 105.24 ^d^	16.82 ± 0.12 ^c^	22.48 ± 0.46 ^e^	16.46 ± 0.13 ^b^	181.58 ± 2.20 ^c^	11.58 ± 0.19 ^c.d^	38.36 ± 2.43 ^b^
12	2826.37 ± 63.43 ^i^	9.48 ± 0.30 ^e^	3.08 ± 0.06 ^h^	9.53 ± 0.49 ^e.f.g.h^	83.55 ± 3.31 ^h^	2.48 ± 0.46 ^g^	19.76 ± 1.39 ^f^
13–15 ^a^	3848.6 ± 157.25 ^h^	11.97 ± 0.29 ^d^	48.77 ± 1.00 ^a^	8.37 ± 0.07 ^g.h^	113.37 ± 4.75 ^e.f^	23.58 ± 0.54 ^a^	27.22 ± 0.66 ^d.e^

^a^ Mean ± standard deviation (n = 3). Means followed by the same letters (in each column) did not differ statistically (*p* > 0.05).

**Table 6 molecules-28-01030-t006:** Box–Behnken experimental design of a three-variable system.

Factors	Symbol	Unit	Levels		
			−1	0	+1
Temperature	T	K	303.15	318.15	333.15
Pressure	P	MPa	15	20	25
Flow rate	F	g.min^−1^	2	3.5	5

## Data Availability

Not applicable.

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
