# Peer review of "Optimization Studies and Compositional Oil Analysis of Pequi (Caryocar brasiliense Cambess) Almonds by Supercritical CO2 Extraction"

_molecules, 2023, doi:10.3390/molecules28031030_

Round 1

Reviewer 1 Report

Optimization studies and compositional oil analysis from  pequi (Caryocar brasiliense Cambess) almonds by supercritical  CO2 extraction study did by author is quite interesting but there is no novelty. Already stablished method they use and extract the fatty acid and oil. This study should  give some  biological applications of extracted oil. This study wanted to emphasis the advantages of super critical extraction but its already known to all, only changing the parameter is not mean any novelty. The author should compare also other extraction method using same plant. Hence, I cant recommend the articles in Molesules.

Author Response

Answer: Oil pequi (Caryocar brasiliense Cambess) is more appropriate to promote significant health benefits and is the primary motivation of this study. Maybe, in Another opportunity, it can satisfy the reviewer by giving some biological applications of extracted oil. The request for extensive English language and style editing is not scored exactly by the reviewer. The submitted manuscript was previously sent to a professional language service. Please see the certificate in the attachment.

Reviewer 2 Report

The manuscript "Optimization Studies and Compositional Oil Analysis from Pequi (Caryocar Brasiliense Cambesss) Almonds by Supercritical CO2 Extraction" shows alternatives to the process of extraction of Pequi oil. However, the authors must clarify the following point:

Pequi oil in addition to the aforementioned components also presents polyphenolic compounds. But the authors did not report it, I would like to know what their conclusion is about the absence of this type of compounds. The temperature used (318.15 K = 45ºC) could be a factor for the denaturation of this type of compounds?

Author Response

We agree as a reviewer about the present polyphenolic compounds in Pequi oil. However, this study focused on the total yield, fatty acid profile, and bioactive compound content. Therefore, the polyphenolic compounds were not investigated. The bioactive compounds identified in Pequi oil do not undergo denaturation due to fast time extraction.

Reviewer 3 Report

1. In the abstract, the “3, 5 and…” (line 18) should be corrected, the number should be 3.5. please correct it.

2. Please adjust the format, lines 33-34.

3. Please add the corresponding references on lines 35-36.

4.The size of the words is different between lines 84-92 and lines 94-97. Please adjust it.

5.Why is there no analysis of the influence of “the square value of each factor” on the yield in table 2?

6. In figure 2C, the letter C should not cover the Figure/picture. In addition, the description of Figure 2C is wrong, please correct it. In Figure 2C, the temperature should be corrected to pressure. Line 111.

7. How do you choose the level of each factor? Is there any basis? Please explain clearly in the method. Based on the Figure 2, the level seems unreasonable.

8. Figure S1 in the supplementary, the number 10 should be instructed to the corresponding device by using a short-line.

Author Response

1. In the abstract, the “3, 5 and…” (line 18) should be corrected, the number should be 3.5. please correct it.

Ans.: We agree. The sentence was corrected.

2. Please adjust the format, lines 33-34.

Ans.: We agree. The adjust the format of the sentence was done.

3. Please add the corresponding references on lines 35-36.

Ans.: We agree. 

4.The size of the words is different between lines 84-92 and lines 94-97. Please adjust it.

Ans.: We agree. The adjust it was done.

5. Why is there no analysis of the influence of “the square value of each factor” on the yield in table 2?

Ans.: The experimental design does not recommend the square effect of factors evaluated.

6. In figure 2C, the letter C should not cover the Figure/picture. In addition, the description of Figure 2C is wrong, please correct it. In Figure 2C, the temperature should be corrected to pressure. Line 111.

Ans.: We agree. The letters in Figure 2 were changed. Line 111 was corrected.

7. How do you choose the level of each factor? Is there any basis? Please explain clearly in the method. Based on the Figure 2, the level seems unreasonable.

Ans.: Generically, the levels of each factor are determined from the research group's experience owned,  in research on the literature about profile chemical of oil similar and behavior phase equilibrium when available. 

8. Figure S1 in the supplementary, the number 10 should be instructed to the corresponding device by using a short-line.

Ans.: We agree.

Reviewer 4 Report

I'd like to congratulate the Authors on this perfect research article which was a pleasure to read. 

Please carefully check the formatting of the text in Line 94-97.

Personally, I prefere graphics over tables. Think about changing table 4 and 5 for graphs and move the tables to the Supplementary Information (just personal preference and suggestion)

All the best and thanks for this nice read before christmas!

Author Response

Please carefully check the formatting of the text in Line 94-97.
Ans.: We agree. The formatted text was adjusted.

Personally, I prefere graphics over tables. Think about changing table 4 and 5 for graphs and move the tables to the Supplementary Information (just personal preference and suggestion)

Ans.: Thank you suggestion, but the graphics and tables in the manuscript permit a reader to read more quickly.

Round 2

Reviewer 1 Report

The manuscript is edited by native speker and now sounds good. Can be accept after careful revision.

Author Response

"The manuscript is edited by native speker and now sounds good. Can be accept after careful revision."

Ans.: Thank you for your time and attention.

Reviewer 3 Report

line 110, temperature, why not correct?

Author Response

"line 110, temperature, why not correct?"

Ans.: We agree. Figure 2C label has been corrected.
